# TYMV and TRV infect *Arabidopsis thaliana* by expressing weak suppressors of RNA silencing and inducing host RNASE THREE LIKE1

Hayat Sehki[1,2], Agnès Yu[1], Taline Elmayan[1], Hervé Vaucheret[1]*

1 Institut Jean-Pierre Bourgin, INRAE, AgroParisTech, Université Paris-Saclay, Versailles, France,
2 Université Paris-Sud, Université Paris-Saclay, Orsay, France

* herve.vaucheret@inrae.fr

**Data Availability Statement:** All relevant data are within the manuscript and its Supporting information files.

## Abstract

Post-Transcriptional Gene Silencing (PTGS) is a defense mechanism that targets invading nucleic acids of endogenous (transposons) or exogenous (pathogens, transgenes) origins. During plant infection by viruses, virus-derived primary siRNAs target viral RNAs, resulting in both destruction of single-stranded viral RNAs (execution step) and production of secondary siRNAs (amplification step), which maximizes the plant defense. As a counter-defense, viruses express proteins referred to as Viral Suppressor of RNA silencing (VSR). Some viruses express VSRs that totally inhibit PTGS, whereas other viruses express VSRs that have limited effect. Here we show that infection with the *Turnip yellow mosaic virus* (TYMV) is enhanced in Arabidopsis *ago1*, *ago2* and *dcl4* mutants, which are impaired in the execution of PTGS, but not in *dcl2*, *rdr1* and *rdr6* mutants, which are impaired in the amplification of PTGS. Consistently, we show that the TYMV VSR P69 localizes in siRNA-bodies, which are the site of production of secondary siRNAs, and limits PTGS amplification. Moreover, TYMV induces the production of the host enzyme RNASE THREE-LIKE 1 (RTL1) to further reduce siRNA accumulation. Infection with the *Tobacco rattle virus* (TRV), which also encodes a VSR limiting PTGS amplification, induces RTL1 as well to reduce siRNA accumulation and promote infection. Together, these results suggest that RTL1 could be considered as a host susceptibility gene that is induced by viruses as a strategy to further limit the plant PTGS defense when VSRs are insufficient.

## Author summary

RNA silencing is a conserved defense mechanism directed against viruses in various eukaryotic kingdoms. As a counter-defense, viruses generally express proteins referred to as viral suppressor of RNA silencing (VSR), which promote infection by inhibiting one or the other component of the RNA silencing machinery. So far, most of the work on VSRs has concentrated on those that strongly inhibit RNA silencing, causing severe infections and plant death. However, situations where VSRs only partially inhibit RNA silencing could be considered as advantageous for both partners of the infection because infected plants survive, flower and produce seeds despite virus multiplication. In this study, we

**Funding:** This work was supported in part by a grant from the French National Research Agency (reference ANR-20-CE12-0025-03) to HV. The funders had no role in study design, data collection and analysis, decision to publish, or preparation of the manuscript.

**Competing interests:** The authors declare that they have no competing interests.

show that *Turnip yellow mosaic virus* (TYMV) encodes a weak VSR, P69, which partially inhibits the amplification but not the execution of RNA silencing. In addition, TYMV induces the expression of the endogenous enzyme RNASE THREE-LIKE 1 (RTL1) to further reduce siRNA accumulation. Similarly, *Tobacco rattle virus* (TRV), which also encodes a weak VSR, induces RTL1 to reduce siRNA accumulation and promote infection. We propose that the limited effect of some VSRs on RNA silencing together with the ability of the corresponding viruses to induce the host RTL1 results in a tight balance between virus propagation and plant development, allowing a virus to propagate without killing its host. In the light of these results, RTL1 could be considered as a susceptibility gene induced by viruses encoding weak VSRs.

## Introduction

As sessile organisms, plants have developed adaptive mechanisms to rapidly cope with biotic (herbivores, pathogens) or abiotic (light, temperature, nutrient, water...) stresses in a fluctuating environment. In the case of pathogen attacks, plants have co-evolved with their pathogens and have put in place a large set of tools to fight against the intruder. One layer of this complex defense involves a process called Post-Transcriptional Gene Silencing (PTGS) [1–3].

Most plant virus belongs to class IV, i.e. their genetic material consists in single-strand RNA (ssRNA) molecule(s). These viruses replicate using their own RNA-dependent RNA Polymerase (RdRP) to form a double-strand RNA (dsRNA) intermediate. In some cases, viral dsRNA can also result from the partial folding of viral ssRNA. Viruses also sometimes produce ssRNA that differ from endogenous mRNA in their structure, lacking either a cap or a polyA tail. Recognized as aberrant RNAs (abRNA), they should be degraded by the cellular RNA quality control (RQC) pathway. However, their amount is probably excessive for the RQC capacity, allowing their transformation into dsRNA by cellular RNA-dependent RNA POLYMERASE (RDR) enzymes, in particular RDR6, which exhibit high affinity for these types of abRNAs. In any case, viral dsRNA molecules are perfect substrates for plant RNase type III enzymes called DICER-LIKE (DCL). In particular, DCL4 and DCL2 cut viral dsRNA into 21- and 22-nt small interfering RNA (siRNA) duplexes, respectively [4–6]. Those duplexes are methylated by HEN1 in their overhang 3' extremities to protect it from uridylation and degradation. Viral siRNA duplexes are loaded onto proteins from the ARGONAUTE (AGO) family, mostly AGO1, but also AGO2, AGO5, AGO7 and AGO10 [7]. The passenger strand of the siRNA duplex is cleaved and eliminated, allowing the annealing of the guide siRNA strand with complementary viral ssRNA molecules and their cleavage owing to the RNaseH activity of AGO proteins. After cleavage by AGO/siRNA complexes, viral ssRNA fragments are degraded by exonucleases. In addition, RNAs targeted by an AGO1/22-nt siRNA complex can attract components of the PTGS amplification machinery, including RDR6, SDE5 and SGS3, allowing their transformation into dsRNA, leading to the production of secondary siRNAs and their subsequent loading onto AGO proteins, which maximizes the elimination of viral RNAs from the plant cell [8–10].

Whereas the PTGS process should eliminate every virus by producing siRNAs from viral dsRNA and returning them against viral ssRNA, examples of plants that actually recover from virus infection are scarce [1]. In fact, most viruses succeed in infecting plants because they encode proteins called VSR, which have the capacity to inhibit PTGS at one or the other step

[2,3,11]. Depending on which PTGS component is targeted by a VSR, PTGS is more or less inhibited, resulting in variable amounts of viruses and symptoms that range between severe and mild. For example, the HC-Pro protein produced by the *Turnip mosaic virus* (TuMV, a member of the Potyviridae family) or the P19 protein produced by the *Tomato bushy stunt virus* (TBSV, a member of the Tombusviridae family) sequester siRNAs, thus preventing their loading onto AGO proteins [2,3,11]. P38, the VSR encoded by the *Turnip crinckle virus* (TCV, another member of the Tombusviridae family) or 2b, the VSR encoded by the *Cucumber mosaic virus* (CMV, a member of the Bromoviridae family), inactivate AGO1 activity [2,3,11]. As a result, CMV, TBSV, TCV and TuMV cause severe symptoms because their VSRs totally block the execution of PTGS. Consistently, mutants impaired in PTGS are as sensitive as wild-type plants to infection by such viruses because PTGS is totally inefficient at counteracting these viruses [10,12–14]. On the other hand, some viruses cause mild symptoms, for example *Turnip yellow mosaic virus* (TYMV), a member of the Tymoviridae family, or *Tobacco rattle virus* (TRV), a member of the Virgaviridae family, suggesting that PTGS is active against such virus. Consistently, mutants impaired in PTGS accumulate more TRV or TYMV RNAs than wild-type plants [15,16]. The fact that PTGS is active against these viruses was explained by the incapacity of their VSRs to totally suppress PTGS. In the case of TRV, its VSR, called 16K, was shown to only affect the amplification step of PTGS [17]. However, in the case of TYMV, the effect of its VSR, called P69, was not fully understood. One report [18] revealed that P69 inhibits PTGS induced by sense transgenes (S-PTGS) but not PTGS induced by inverted-repeat transgenes (IR-PTGS). These authors also reported that P69 provokes the accumulation of miRNA guide and passenger strands, leading them to propose a mechanism for viral virulence based on miRNA-guided inhibition of host gene expression. Here we investigated further the effect of P69 on PTGS and found that P69 localizes in siRNA-bodies, where it partially inhibits the production of secondary siRNAs, thus limiting PTGS amplification. We also show that TYMV reinforces the limitation of PTGS amplification by inducing the expression of the endogenous enzyme RNASE THREE-LIKE1 (RTL1), which degrades dsRNA precursors of siRNA [19]. Similarly, TRV, which VSR 16K only affects the amplification step of PTGS [17], induces RTL1, which further contributes to limiting the accumulation of antiviral siRNAs. RTL1 could therefore be considered as a susceptibility gene induced by viruses that have limited effect of the plant PTGS defense.

## Results

### Identification of PTGS mutations that aggravate symptoms of TYMV infection

A previous study revealed that *dcl4* mutants infected with TYMV exhibit aggravated symptoms and accumulate twice as much viral RNA compared to wild-type plants, suggesting that TYMV RNAs are targeted by PTGS [16]. This result was somehow surprising considering that previous analyses revealed that *dcl2 dcl4* double mutants but not *dcl2* or *dcl4* single mutants were more susceptible to VSR-deficient CMV and TuMV [1–4]. Moreover, aggravated symptoms and increased viral RNA levels were also observed in *ago1 ago2* and *rdr1 rdr6* doubles mutants but not in the corresponding single mutants infected with VSR-deficient CMV and TuMV [1–4]. Therefore, to further decipher which PTGS components play a role in anti-TYMV PTGS, a series of mutants was infected by TYMV. At first, *ago1*, *ago2*, *dcl2*, *dcl4*, *rdr1* and *rdr6* single mutants were infected. *ago1*, *ago2* and *dcl4*, but not *dcl2*, *rdr1*, and *rdr6* exhibited aggravated symptoms, *i.e.* reduced growth and leaf yellowing, when compared with infected Col (Fig 1). Then, *ago1 ago2*, *dcl2 dcl4* and *rdr1 rdr6* doubles mutants were infected. Symptoms were aggravated in *ago1 ago2* compared to *ago1* and *ago2*, but symptoms were

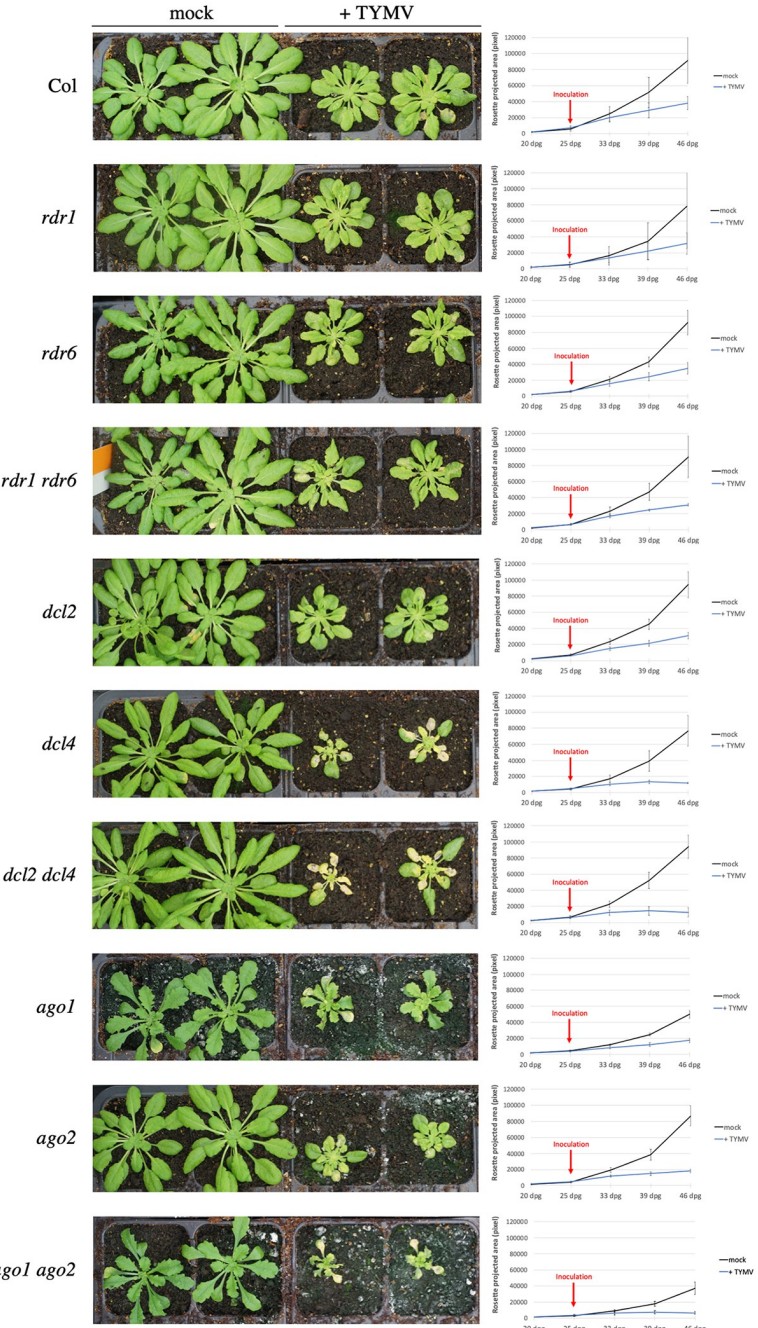

**Fig 1. The plant PTGS defense limits TYMV infection through the action of AGO1, AGO2 and DCL4.** Pictures of mock- and TYMV-infected plants of the indicated genotypes three weeks after infection with TYMV. Plants were grown under short day conditions. The growth curves show the temporal change of rosette area (averaged by four plants, +/-SE) in mock- and TYMV-infected plants from one week pre-inoculation to three weeks post-inoculation. dpg: days post-germination.

unchanged in *dcl2 dcl4* compared to *dcl4*, and in *rdr1 rdr6* compared to Col, *rdr1* and *rdr6* (Fig 1). Quantification of viral RNA confirmed that AGO1, AGO2 and DCL4 contribute to limiting viral RNA levels (S1 Fig). Given that AGO1, AGO2 and DCL4 are sufficient to execute PTGS using primary siRNAs, whereas DCL2, RDR1 and RDR6 are necessary for the amplification of

PTGS, these results suggest that TYMV infection is limited by the action of DCL4-dependent primary siRNAs that are loaded onto AGO1 and AGO2 to execute PTGS. The corollary to this hypothesis is that TYMV eventually infects successfully plants because it inhibits the amplification step of PTGS, making the action of primary siRNAs insufficient to completely prevent infection.

## TYMV inhibits the amplification step of PTGS

To further challenge the hypothesis that TYMV inhibits the amplification step of PTGS, several transgenic lines were infected by TYMV. Lines *L1* and *6b4* carry the same *p35S:GUS-tRbcS* transgene. Line *L1* spontaneously undergoes a form of PTGS referred to as S-PTGS for sense-transgene induced PTGS. On the other hand, line *6b4* stably expresses *GUS* mRNA. However, the *6b4* locus is prone to trigger S-PTGS. Indeed, *6b4 ski3*, *6b4 xrn4* and *6b4 vcs* lines, in which RQC is impaired in either the exosome, XRN or decapping function, trigger S-PTGS [20–22]. This indicates that, in a wild-type background, the aberrant RNAs produced by the *6b4* locus are degraded by RQC, but that these aberrant RNAs can be transformed into dsRNA when RQC is not actively degrading them. Silencing of the *p35S:GUS-tRbcS* transgene carried by the *6b4* locus can also be achieved when the *6b4* locus is brought into the presence of the *306* locus carrying a *p35S:hpG* transgene. The *306* locus [23] produces an hairpin RNA made of the 5' of the *GUS* sequence followed by the 3' end of the *GUS* sequence, itself followed by the 5' of the *GUS* sequence in reverse orientation (Fig 2A). The *306* locus directly produces a dsRNA that is transformed into siRNA by DCL2 and DCL4 without the requirement of any RDR, resulting in the destruction of *GUS* mRNA produced by the *p35S:GUS-tRbcS* transgene of the *6b4* locus. This form of PTGS is referred to as IR-PTGS for inverted repeat-induced PTGS. In wild-type plants, amplification also occurs through RDR6 activity, resulting in the production of secondary siRNAs from the 231 bp fragment of the *GUS* sequence that is present in the *6b4* locus but not in the *306* locus. The production of these secondary siRNAs through PTGS amplification is abolished in *rdr6* mutants, but this does not compromise efficient IR-PTGS of the *6b4* locus because enough primary siRNAs are produced from the *306* locus [23,24].

Silenced lines *L1*, *6b4 ski3*, *6b4 xrn4*, *6b4 vcs* and *6b4-306* were infected with TYMV to determine the impact of TYMV on S-PTGS and IR-PTGS. GUS activity was observed in *L1*, *6b4 ski3*, *6b4 xrn4* and *6b4 vcs*, but not *6b4-306* plants (Fig 2B and 2C), suggesting that TYMV inhibits S-PTGS, which is amplification-dependent, but not IR-PTGS, which is amplification-independent, confirming previous results obtained using different reporters [18].

Given that GUS activity was lower in *L1*-infected plants than in *6b4 ski3*-, *6b4 xrn4*- or *6b4 vcs*-infected plants, we asked whether TYMV-induced suppression of S-PTGS could be more efficient when RQC is abolished. To test this hypothesis, plants carrying the *L1* locus in RQC-deficient mutant backgrounds (*ski3* and *vcs)* were infected with TYMV. GUS activity was similar in infected *L1*, *L1 ski3* and *L1 vcs* (Fig 2C), indicating that TYMV-induced suppression of S-PTGS occurs independently of RQC. Therefore, the stronger suppressing effect of TYMV observed in *6b4 ski3*, *6b4 xrn4*, *6b4 vcs*, compared to *L1*, is more likely due to the fact that S-PTGS is less efficient in *6b4 ski3*, *6b4 xrn4* and *6b4 vcs* than in *L1* [20,22] (Moreno et al, 2013), and thus easier to inhibit.

## The TYMV VSR P69 inhibits the amplification step of PTGS

The TYMV genome encodes four proteins, among which P69 appears necessary for cell-to-cell movement of the virus [25] and for suppression of *p35S:GUS* S-PTGS [18]. However, the experiments originally performed by Chen et al., 2004 were conducted using a *p35S:P69*

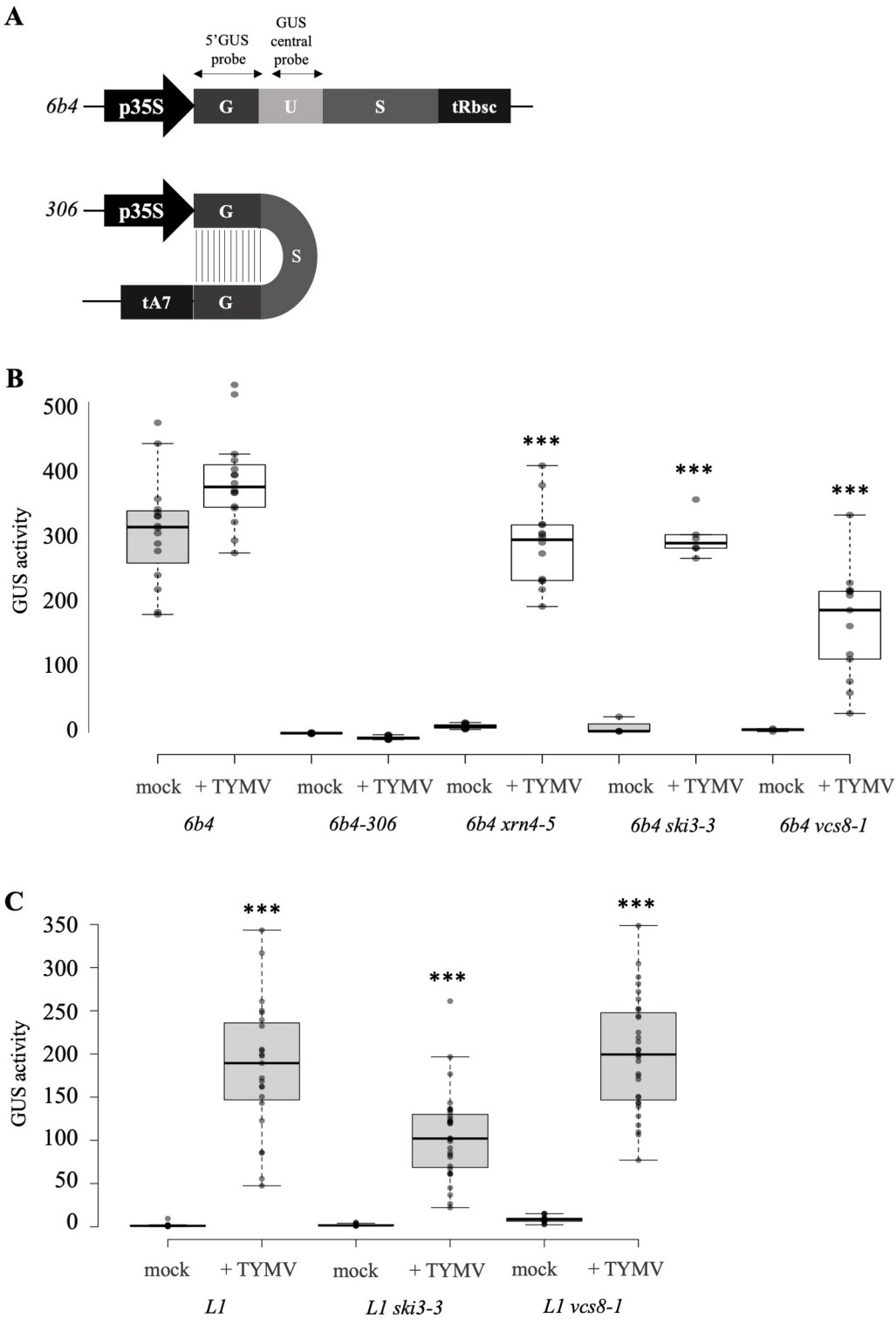

**Fig 2. TYMV inhibits S-PTGS but not IR-PTGS. A)** Scheme of the *p35S:GUS* and *p35S:hpG* in *6b4* and *6b4-306* transgenic lines. Probes used for northern blot revelation are indicated. **B) and C)** GUS activity in transgenic Arabidopsis lines infected with TYMV. Plants were grown in long day conditions. GUS activity was measured three weeks after infection and is expressed in arbitrary unity of fluorescence/ug of protein/minutes. 3 stars indicate a student test with a significant level <0.01 between mock and infected plants.

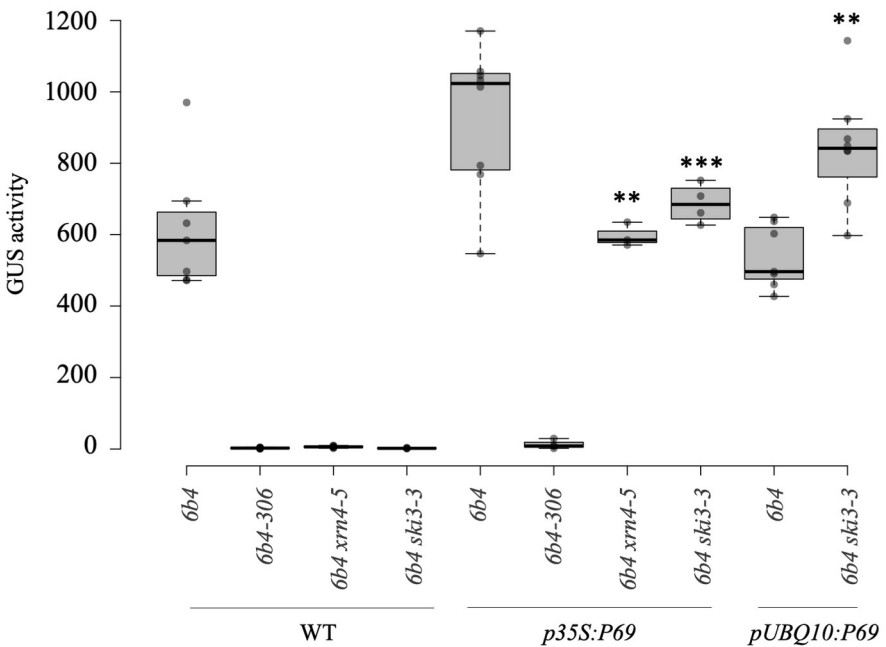

**Fig 3. The TYMV protein P69 inhibits S-PTGS but not IR-PTGS.** GUS activity in control or transgenic lines transformed with a *pUBQ10:P69* or *p35S:P69* construct. GUS activity is expressed in an arbitrary unit of fluorescence/ μg of protein/min. 3 stars indicate a student test with a significant level <0.01 and 2 stars <0.05 between WT and transformed plants.

construct that probably produces P69 above the level existing in TYMV-infected plants. Therefore, lines *6b4 ski3*, *6b4 xrn4* and *6b4-306* were transformed with either a *p35S:P69* or a *pUBQ10:P69* construct. None of the *6b4-306/p35S:P69* or *6b4-306/pUBQ10:P69* plants exhibited GUS activity (Fig 3). In contrast, GUS activity was observed in most *6b4 ski3/p35S:P69*, *6b4 ski3/pUBQ10:P69*, *6b4 xrn4/p35S:P69* and *6b4 xrn4/pUBQ10:P69* plants, indicating that P69 suppresses S-PTGS. To confirm that P69 has no effect on IR-PTGS, the *pUBQ10:P69* and *p35S:P69* transformants expressing the most *P69*, chosen as the *6b4 ski3/pUBQ10:P69* and *6b4 xrn4/p35S:P69* transformants exhibiting the highest GUS activity, were either selfed or crossed to *6b4-306* and GUS activity was measured in F1 plants. GUS activity was observed in selfed *6b4 ski3/pUBQ10:P69* and *6b4 xrn4/p35S:P69* plants but not in *6b4-306* x *6b4 ski3/ pUBQ10:P69* and *6b4-306* x *6b4 xrn4/p35S:P69* plants, confirming that P69 inhibits amplification-dependent S-PTGS but not amplification-independent IR-PTGS.

### The TYMV VSR P69 likely inhibits the production of dsRNA, not its dicing

S-PTGS amplification not only requires the action of an RDR to produce dsRNA, but also depends on the action of DCL2. Indeed, DCL2-dependent 22-nt siRNAs, but not DCL4-dependent 21-nt siRNAs, promote the transformation of targeted ssRNA into dsRNA by RDR6 and the production of secondary siRNAs [8,9,26]. The fact that *dcl2* and *rdr6* are not more susceptible to TYMV than Col indicates that the amplification step is targeted by P69, however it does not say if it is the production of dsRNA or their dicing by DCL2 that is affected by P69. To resolve this question, we examined the accumulation of endogenous siRNAs, which depend either on DCL2 or RDR6 for their production. At first, we examined the accumulation *TAS3* ta-siRNAs, which depend on RDR6 but not DCL2 for their production because it is not

initiated by a cut mediated by a DCL2-dependent 22-nt siRNAs but by a miRNA called miR390. As a result, *rdr6* but not *dcl2* mutants exhibit downward curling of the leaf margin [27]. *pUBQ10:P69* plants also exhibit downward curling of the leaf margins (Fig 4A), and northern blot analysis confirmed that *TAS3* ta-siRNAs accumulate at lower level in *pUBQ10*:*P69* plants compared to wild-type plants, whereas miR390 level remained unchanged (Fig 4B). Then, we examined the accumulation of the endogenous *IR71* 22-nt siRNAs. Their production requires DCL2 but not RDR6 because *IR71* dsRNA is made by internal folding of a long ssRNA. *IR71* 22-nt siRNAs accumulated at similar level in *pUBQ10:P69* and wild-type plants (Fig 4B), indicating that DCL2 action is not impaired by P69.

To confirm that P69 inhibits the RDR6-dependent amplification step, we took advantage of the *6b4-306* line. In this line, the production of secondary siRNAs from the 231 bp fragment of the *GUS* sequence that is present in the *6b4* locus but not the *306* locus (Fig 2A) depends on RDR6 [23]. In *6b4-306/pUBQ10:P69* and *6b4-306/p35S:P69* plants, IR-PTGS is not abolished due to the action of primary siRNAs produced from the *306* locus (Fig 2B), but the production of secondary siRNAs is reduced in *pUBQ10:P69* plants (Fig 4B) and abolished in *p35S:P69* plants (Fig 4C), confirming that the amplification, but not the execution, of S-PTGS is impaired by P69.

## The TYMV VSR P69 localizes in siRNA-bodies where actors of siRNA amplification reside

RDR6 and SGS3, the two major components of S-PTGS amplification, reside in cytoplasmic foci called siRNA-bodies (SB). These foci are very small and almost impossible to detect under normal conditions. However, after heat stress or osmotic stress, larger foci start to appear [21,28–31]. These larger SB resemble Stress Granules (SG), which consist in ribonucleoprotein complexes where mRNAs are stored during stress [32]. After stress, RDR6 and SGS3 colocalize with POLYADENYLATE-BINDING 2 (PAB2), a typical SG marker, suggesting either that SG derive from SB or that SB fuse to SG during stress [21,30].

To get further insight on how P69 inhibits S-PTGS amplification, P69 sub-cellular localization was examined. For this purpose, *pUBQ10:P69-GFP* and *pUBQ10:GFP-P69* constructs were generated and introduced into tobacco leaves by agro-infiltration. A dual cytoplasmic and nuclear localization was observed (Fig 5A), confirming previous results obtained by introduction of a *p35S:P69-GFP* construct into protoplasts [33]. Then, stable *Arabidopsis* transformants carrying *pUBQ10:P69-GFP* were produced. Confocal analyses of roots revealed that these transformants exhibit a diffuse cytoplasmic GFP signal but no clear nuclear signal (Fig 5B). After stress, the GFP signal was observed in foci that resemble SB and SG (Fig 5B).

To determine if P69 localizes in SB and/or SG, *pUBQ10:P69-GFP* plants were crossed to *p35S:SGS3-mCherry* and *pPAB2:PAB2-RFP* plants. Confocal analyses of untreated roots revealed diffuse cytoplasmic GFP, mCherry and RFP signals. However, after heat stress, GFP and mCherry signals colocalized in large foci in *pUBQ10:P69-GFP* x *p35S:SGS3-mCherry* plants, and GFP and RFP signals colocalized in large foci in *pUBQ10:P69-GFP* x *pPAB2*: *PAB2-RFP* plants (Fig 5B), strongly suggesting that P69 resides in SB and/or SG where it somehow limits PTGS amplification.

## RTL1 induction during TYMV infection reinforces the effect of P69

The results described above indicate that the TYMV VSR P69 has the capacity to inhibit the amplification step of the antiviral PTGS. However, the physiological level of P69 during infection appears insufficient to totally inhibit this step, which only appears possible when P69 is expressed constitutively from a transgene (compare Figs 2B and 3) and at high level (compare

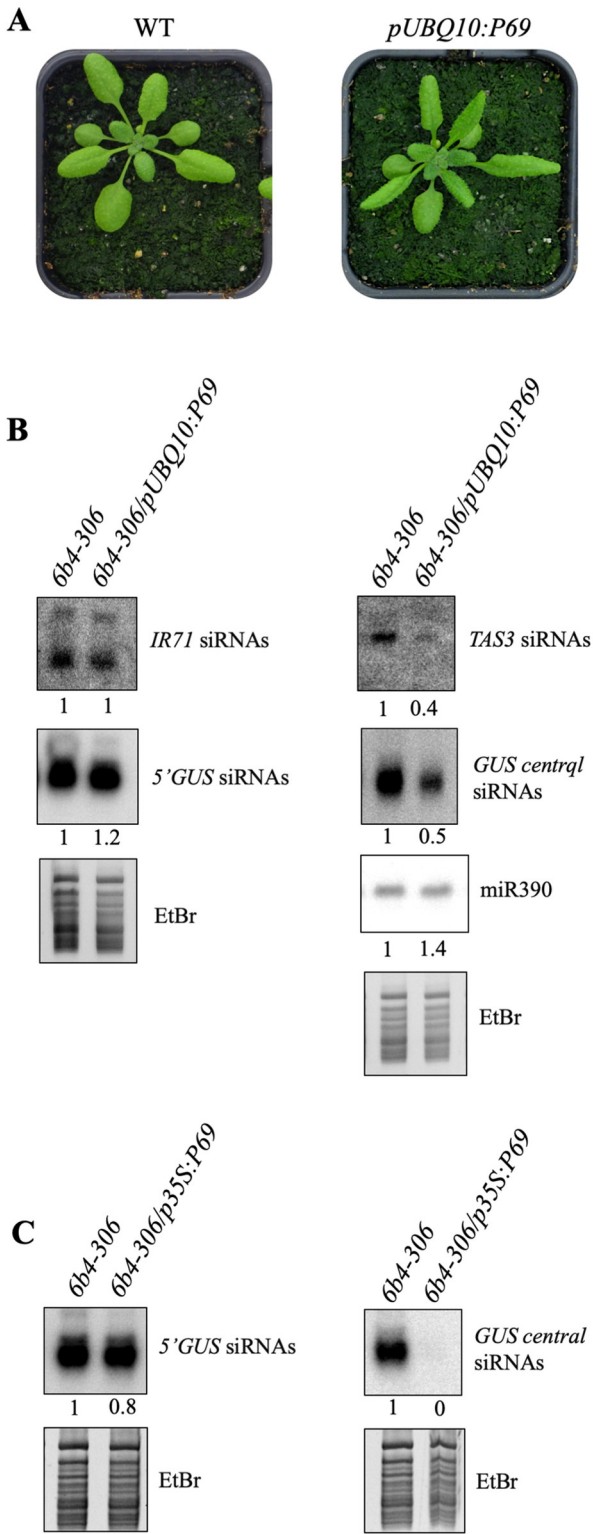

**Fig 4. The TYMV protein P69 inhibits the production of PTGS secondary siRNAs. A)** Phenotype of wild-type plants and transgenic lines carrying a *pUBQ10:P69* construct. Downward leaf curling is typical of the absence of *TAS3* siRNAs. **B)** Accumulation of miR390, *TAS3* and *IR71* endogenous siRNAs and *GUS* primary and secondary siRNAs (using 5'GUS and central GUS probes, respectively, see Fig 2A) in control or transgenic plants transformed with the *pUBQ10:P69* construct. EtBr staining is shown as loading control. **C)** Accumulation of *GUS* primary and secondary

siRNAs in control or transgenic plants transformed with the *35S:P69* construct. EtBr staining is shown as loading control. siRNA accumulation is resumed as band intensity measured with ImageJ and normalized to total RNA bands intensity on EtBr staining.

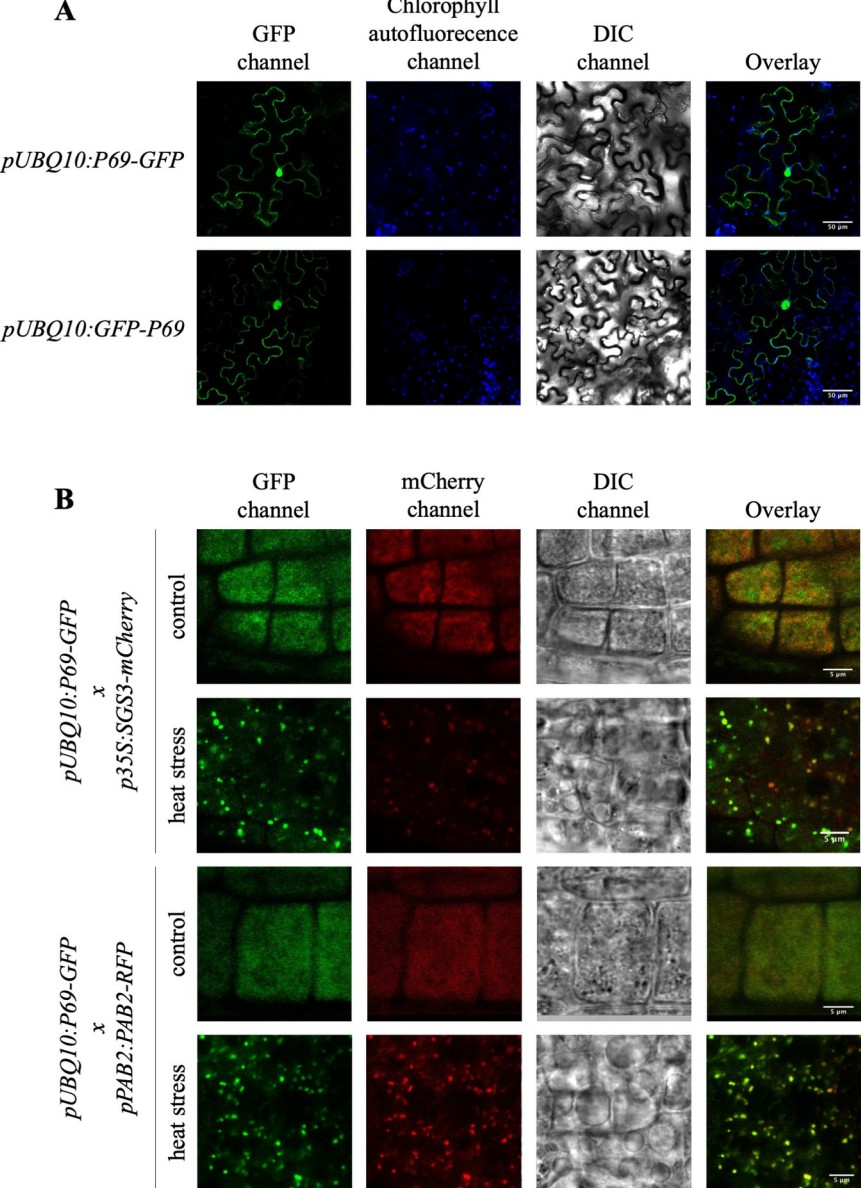

**Fig 5. The TYMV protein P69 colocalizes with SB and/or SG under stress condition in *Arabidopsis thaliana*. A)** *Nicotiana benthamiana* leaves agroinfiltrated with *pUBQ10:P69-GFP* or *pUBQ10:GFP-P69* constructs. GFP (green) and chlorophyll auto-fluorescence (blue) signals were analyzed by confocal microscopy 2 days post-agroinfiltration. The overlay corresponds to the merge between the two signals. Channels are indicated above each column and scale bars (50 μm) are indicated on the overlay. DIC: Differential Interference Contrast. **B)** Subcellular localization of GFP, RFP and mCherry determined before and after a heat stress of 1h at 37°C in 5 days old transgenic *Arabidopsis thaliana* lines *pUBQ10:P69-GFP* crossed with *p35S:SGS3-mCherry* or *pPAB2:PAB2-RFP* lines. The overlay corresponds to the merge between the GFP and the RFP or mCherry signals. Channels are indicated above each column and scale bars (5 μm) are indicated on the overlay. DIC: Differential Interference Contrast.

Fig 4B and 4C). Given that TYMV successfully infect Arabidopsis, we asked if TYMV could take advantage of the endogenous RNASE-THREE-LIKE1 (RTL1) to achieve its infection. Indeed, we previously reported that plants constitutively expressing *RTL1* (*p35S:RTL1*) exhibit aggravated symptoms and accumulate more viral RNA than wild-type plants when they are infected with TYMV [19]. RTL1 is an RNaseIII enzyme that cleaves perfectly paired dsRNA of endogenous or exogenous origin, thus preventing the production of siRNAs, including RDR6-dependent siRNAs [5]. The endogenous *RTL1* gene is not expressed in Arabidopsis vegetative tissues, but is induced by various types of viruses, suggesting that *RTL1* induction is a general response to virus infection. However, most viruses encode a VSR that suppresses RTL1 activity. TYMV sets apart because its VSR P69 does not inhibit RTL1 activity [5]. To address whether *RTL1* induction during TYMV infection could reinforce the effect of P69 on PTGS amplification, the sensitivity of *rtl1* mutants to TYMV infection was analyzed. One T-DNA mutant, referred to as *rtl1-1*, was identified in the SAIL collection. This mutant carries an insertion in the middle of the RNaseIII domain (Figs 6A and S2), thus abolishing RTL1 activity. The second mutant, referred to as *rtl1-2*, was obtained using the CRISPR-Cas9 technology. A one-base insertion at base 27 after the ATG causes a frameshift and the production of a truncated protein of only 11 amino acids, which lacks both RNaseIII and DRB domains (Figs 6A and S2). These two mutants were back-crossed six times to Col to ensure the elimination of unlinked mutations, and homozygous *RTL1/RTL1* and *rtl1/rtl1* siblings were identified after selfing. Then, WT plants, *rtl1* mutants and a *p35S:RTL1* line were infected mechanically with TYMV. The *p35S:RTL1* line exhibited increased symptoms compared with Col and *rtl1* mutants (Fig 6B). Compared to Col plants, *rtl1* mutants exhibited a higher level of TYMV siRNAs and a lower level of TYMV genomic RNA (gRNA), whereas *p35S:RTL1* plants exhibited a lower level of TYMV siRNAs and a higher level of TYMV gRNA (Fig 6C and 6D). Together, these results indicate that RTL1 favors TYMV infection not only when expressed artificially at high level using a *p35S:RTL1* transgene but also when expressed at physiological level during infection of WT plants. They also suggest that TYMV infection is successful due to the dual effect of P69 and RTL1 on PTGS.

Because a fraction of the seeds harvested on TYMV-infected Arabidopsis plants transmit the virus and develop symptoms similar to those observed after mechanical infection [34], we investigated whether the *rtl1* mutation could affect the frequency of TYMV transmission through seeds. 396 seeds harvested on Col-infected plants and 396 seeds harvested on *rtl1-1*-infected plants were sown on soil, and the number of plants developing TYMV symptoms was scored. No difference in the frequency of infected plants was observed between Col and *rtl1-1* (18% in each case), indicating that RTL1 has no effect on virus transmission through seeds. The amount of TYMV siRNA was also monitored in Col and *rtl1-1* plants that developed TYMV symptoms. Similar to what was observed for mechanically infected plants (Fig 6D), a higher level of TYMV siRNAs was observed in the *rtl1-1* plants that transmitted the virus through seeds compared with Col plants that transmitted the virus through seeds (Fig 6E), confirming that RTL1 actually limits the production of antiviral siRNAs in vegetative tissues.

## Counteracting the PTGS defense through RTL1 induction is a strategy also used by TRV

To determine if the RTL1 induction strategy used by TYMV to promote infection can be generalized to other viruses encoding VSR that are not capable of blocking PTGS execution, the relationship between TRV, PTGS and RTL1 was examined. TRV was chosen because like TYMV, TRV causes mild symptoms on Arabidopsis, suggesting that PTGS efficiently limits TRV infection. Supporting this hypothesis, the TRV protein 16K was previously identified as a

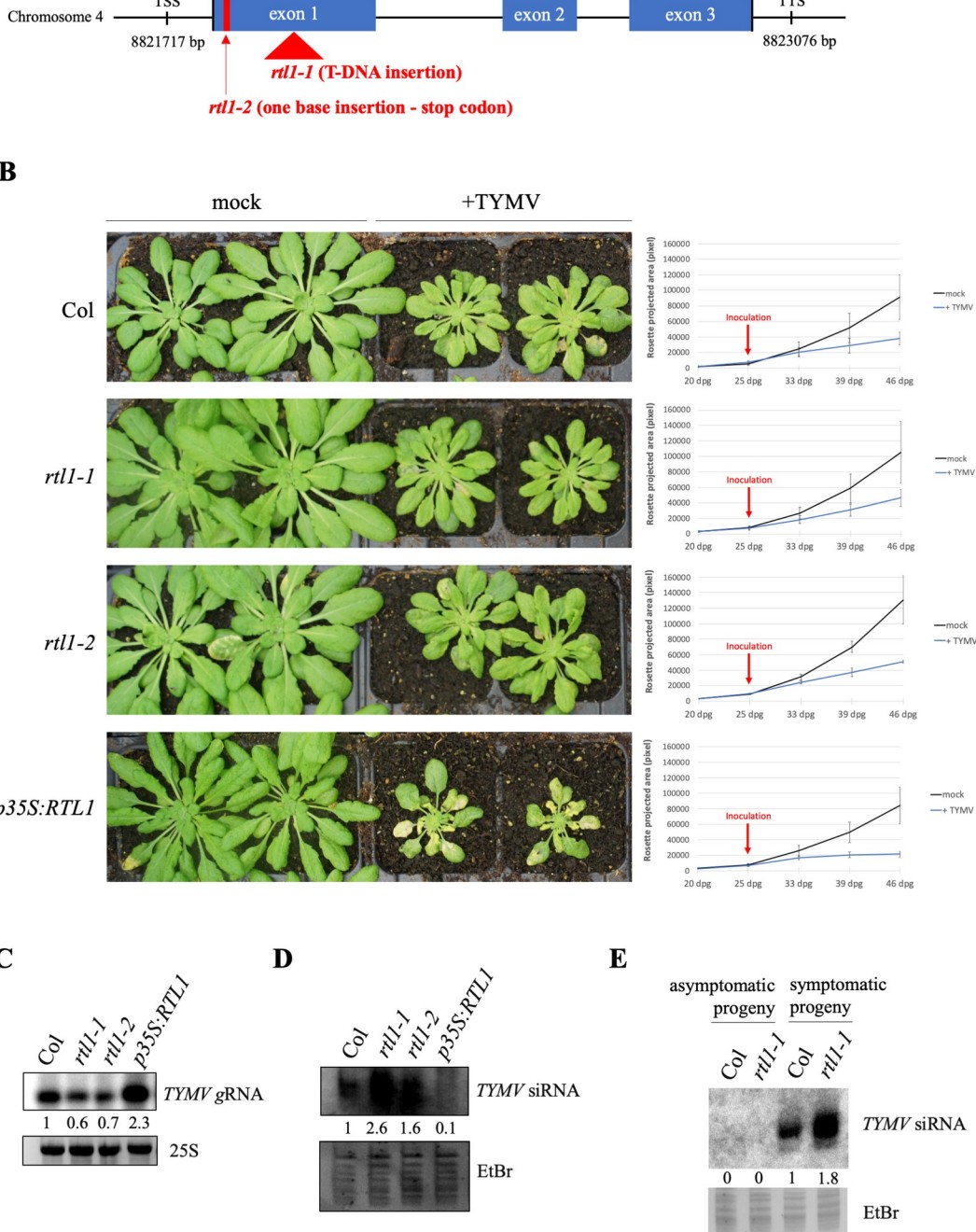

**Fig 6. RTL1 favors TYMV infection by counteracting anti-viral PTGS. A)** Scheme of the *RTL1* gene with its coordinates on chromosome 4. The location of the *rtl1-1* mutation (T-DNA insertion SAILseq_337_F04.1) and of the *rtl1-2* mutation (CRISPR-induced one-base insertion) are indicated. TSS: Transcription Start Site, TTS: Transcription Termination Site. **B)** Pictures of wild type, *rtl1-1* and *rtl1-2* mutants and *p35S:RTL1* plants three weeks after infection with TYMV. Plants were grown under short day conditions. The growth curves show the temporal change of rosette area (averaged by four plants, +/-SE) in mock- and TYMV-infected plants from one week pre-inoculation to three weeks post-inoculation. dpg: days post-germination. **C)** TYMV full-length genomic RNA (gRNA) accumulation in *Arabidopsis thaliana* wild type, *rtl1-1* and *rtl1-2* mutants and *p35S:RTL1* plants four weeks after infection with TYMV. Total RNA was extracted from a pool of 16 infected plants per genotype, run onto an agarose gel and hybridized with a TYMV probe. Data are normalized to Col. **D)** TYMV siRNA accumulation in *Arabidopsis thaliana* wild type, *rtl1-1* and *rtl1-2* mutants and *p35S:RTL1* plants four weeks after

infection with TYMV. Total RNA was extracted from a pool of 16 infected plants per genotype, run onto an acrylamide gel and hybridized with a TYMV probe. Data are normalized to Col. **E)** TYMV siRNA accumulation in asymptomatic *vs* symptomatic progeny of *Arabidopsis thaliana* wild type and *rtl1-1* mutants infected with TYMV. Total RNA was extracted from a pool of 8 plants of each type, run onto an acrylamide gel and hybridized with a TYMV probe. Data are normalized to infected Col.

VSR having limited activity on PTGS amplification [17]. The TRV protein 29K was also proposed to act as a VSR, but its activity was even lower than that of 16K [17,35].

To determine if RTL1 plays a role in TRV infection, we first examined if *RTL1* was induced during TRV infection. Results indicate that like other viruses, TRV induces *RTL1* expression (Fig 7A). Then, wild-type plants, *rtl1* mutants and *p35S:RTL1* plants were challenged with TRV. The *dcl2 dcl4* double mutant was used as a control because it was previously shown to be hypersusceptible to TRV infection (Donaire et al., 2008) [15]. Similar to the *dcl2 dcl4* double mutant, *p35S:RTL1* plants exhibited aggravated symptoms (Fig 7B), which correlated with a higher viral gRNA level (Fig 7C) and the absence of 21-22-nt viral siRNA (Fig 7D). These results confirm that PTGS actually strongly limits TRV infection and indicate that PTGS action can be erased when over-expressing *RTL1*. In contrast to *p35S:RTL1* plants, *rtl1* mutants accumulated more viral siRNAs than wild-type plants (Fig 7D), indicating that the induction of endogenous *RTL1* during TRV infection (Fig 7A) actually limits PTGS activity. This increased level of viral siRNAs in *rtl1* mutants correlated with a slightly reduced level of viral gRNA (Fig 7C).

## Discussion

The outcome of virus infection is determined by the balance between the virus attacks, the host defenses and the virus counter-defenses. In plants, PTGS acts as a sequence-specific defense mechanism directed against viruses. PTGS is activated by dsRNA intermediates of viral replication and/or folded viral RNAs, leading to the production of primary siRNAs that target viral ssRNA for destruction and production of secondary siRNAs that maximize the plant defense. However, most viruses have evolved counter-defenses based on viral proteins called VSR, which inhibit one or the other step of PTGS, thus impacting the plant PTGS defense. VSRs that target essential steps of PTGS execution, for example by sequestrating siRNAs or inhibiting AGO-guided cleavage of viral RNAs, cause very severe symptoms because the plant PTGS defense is totally inhibited. As a result, PTGS-deficient mutants and wild-type plants are similarly infected by the corresponding viruses, and only the use of VSR-deficient viruses allows revealing the capacity of PTGS to actually eliminate these viral RNA molecules [10,12–14]. However, not every virus encodes a VSR capable of totally inhibiting PTGS. As a result, several viruses only provoke mild symptoms on wild-type plants because PTGS remains capable of reducing the amount of viral RNA.

Here, we examined the case of one such virus, the TYMV. The potent action of PTGS against TYMV was revealed by the increased level of TYMV RNA in Arabidopsis *dcl4* mutants [16], indicating that PTGS is at work against TYMV. Nevertheless, the fact that wild-type plants do not fully recover from TYMV infection indicates that PTGS is not capable to eliminate all viral RNAs, suggesting that at least one PTGS step is inhibited by the TYMV. Investigating the behavior of various PTGS mutants infected by TYMV revealed that *dcl4*, *ago1* and *ago2* exhibit enhanced symptoms compared with wild-type plants, whereas *dcl2*, *rdr1* and *rdr6* exhibit symptoms similar to wild-type plants (Fig 1). Because AGO1, AGO2 and DCL4 are involved in the execution of PTGS, whereas DCL2, RDR1 and RDR6 are dispensable for PTGS execution and only contribute to PTGS amplification, these results suggested that only the

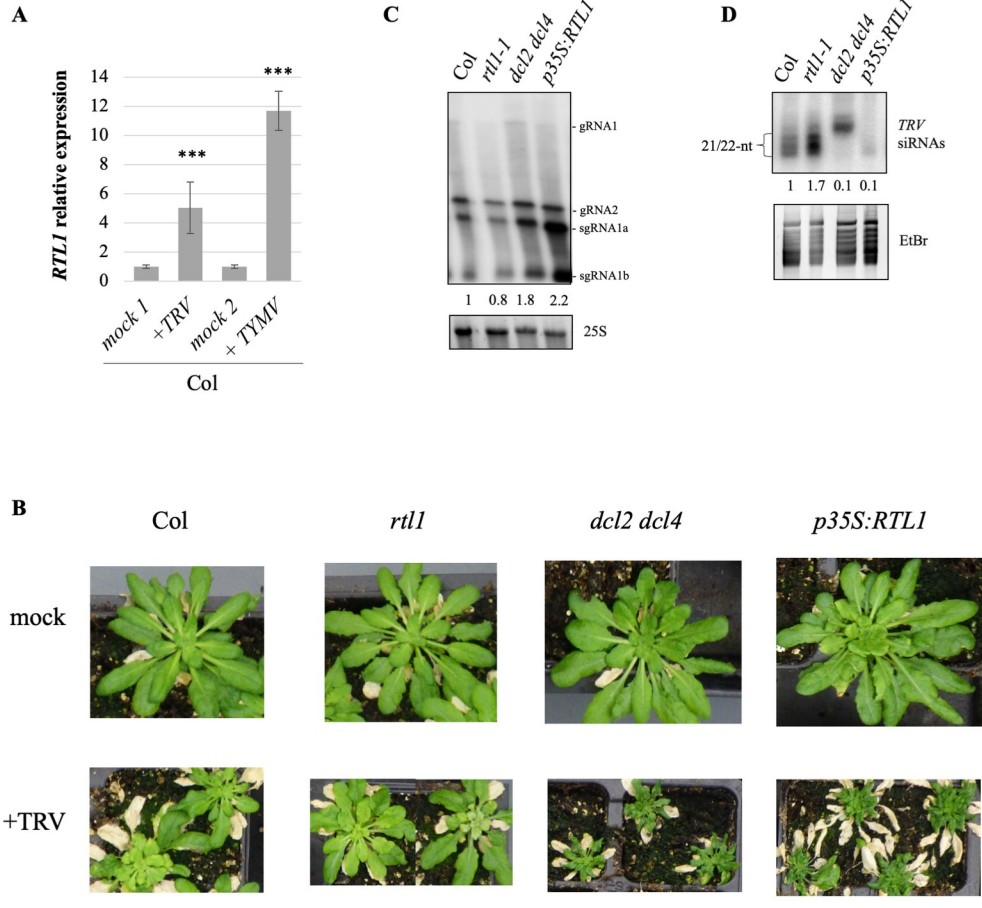

**Fig 7. RTL1 favors TRV infection by counteracting anti-viral PTGS. A)** *RTL1* induction by TRV compared to TYMV. *RTL1* expression was analyzed by qRT-PCR on WT plants three weeks after infection. *GAPDH* expression is used as internal control and data are normalized to non-infected (mock) plants. **B)** Pictures of *Arabidopsis thaliana* wild type, *rtl1-1* and *dcl2 dcl4* mutants and *p35S:RTL1* plants three weeks after infection with TRV. Plants were grown under short day conditions.**C)** TRV genomic RNAs (gRNA) accumulation in *Arabidopsis thaliana* wild type, *rtl1-1* and *dcl2 dcl4* mutants and *p35S:RTL1* plants four weeks after infection with TRV. Total RNA was extracted from a pool of 16 infected plants per genotype and hybridized with a TRV probe. Data are normalized to Col.**D)** TRV siRNA accumulation in *Arabidopsis thaliana* wild type, *rtl1-1* and *dcl2 dcl4* mutants and *p35S:RTL1* plants four weeks after infection with TYMV. Total RNA was extracted from a pool of 16 infected plants per genotype and hybridized with a TRV probe. Data are normalized to Col. For siRNA quantification, only 21 and 22-nt bands were considered.

amplification step of PTGS was impaired during TYMV infection. This hypothesis was confirmed by infecting Arabidopsis lines carrying reporter transgenes silenced either by S-PTGS, which requires PTGS amplification components, or IR-PTGS, which does not require PTGS amplification components. TYMV infection suppressed S-PTGS but not IR-PTGS (Fig 2), and this suppression could be recapitulated by expressing the TYMV VSR P69 only (Fig 3). Moreover, the accumulation of endogenous ta-siRNAs, which production requires RDR6 but not DCL2, was impacted in plants expressing the TYMV VSR P69, whereas the accumulation of endogenous endo-siRNAs, which production requires DCL2 but not RDR6, was not impacted in plants expressing the TYMV VSR P69 (Fig 4). Finally, P69 was shown to localize in siRNA-bodies where RDR6 resides and where PTGS amplification occurs (Fig 5), indicating that P69 limits the PTGS amplification step occurring in siRNA-bodies, thus reducing the production of secondary siRNAs. Therefore, P69 can be added to the list of VSRs that inhibit PTGS

amplification, which includes: i) the VSR protein V2 encoded by the DNA virus *Tomato yellow leaf curl virus* (TYLCV), which competes with the tomato homologue of the Arabidopsis RDR6 cofactor SGS3 for binding to dsRNA [29, 31], ii) the VSR protein TGBp1 encoded by the *Plantago asiatica mosaic virus* (PlAMV), which interacts with SGS3 and RDR6 and coaggregates with SGS3/RDR6 bodies [36], iii) the VSR protein P6 encoded by the *Rice yellow stunt virus* (RYSV), which interacts with RDR6, thus blocking secondary siRNA synthesis [37], iv) the VSR protein βC1 encoded by the *Tomato yellow leaf curl Chinavirus* (TYLCCNV) DNA satellite, which interacts with the endogenous suppressor of silencing calmodulin-like protein (rgs-CAM) *in N. benthamiana* to repress RDR6 expression and secondary siRNA production [38], v) the VSR protein Pns10 encoded by the *Rice dwarf phytoreovirus* (RDV), which downregulates RDR6 [39], and vi) the VSR protein 16K encoded by the *Tobacco rattle virus* (TRV), which somehow limits PTGS amplification [17].

The infectious capacity of viruses expressing VSRs that only partially inhibit the amplification step of PTGS suggested that such viruses could use additional strategies to counteract the plant PTGS defense. This prompted us to investigate whether RTL1 could contribute to reducing the amounts of antiviral siRNAs. *RTL1* is naturally not expressed in wild-type Arabidopsis plants, but is induced following virus infection, suggesting that *RTL1* induction could be an alternative, although not exclusive, way used by viruses to counteract the plant PTGS defense in addition to expressing VSRs, in particular when expressing VSRs that only partially inhibit PTGS. Supporting this hypothesis, *p35S:RTL1* Arabidopsis plants expressing *RTL1* constitutively at high level lacked TYMV siRNAs, accumulated high levels of TYMV gRNA, and exhibited severe symptoms when infected by TYMV [19]. Here, we show that the same holds true for TRV (Fig 7), indicating that, at least in Arabidopsis, RTL1 actually is capable of preventing siRNA-mediated degradation of TRV and TYMV RNAs. However, *p35S:RTL1* plants accumulated RTL1 above the level observed in wild-type infected plants [19], and the demonstration of the role of RTL1 awaited the analysis of *rtl1* mutants. Here we showed that TRV- and TYMV-infected *rtl1* mutants accumulate more TRV or TYMV siRNAs and less TRV or TYMV gRNAs than wild-type plants (Figs 6 and 7). Together, these results suggest a model where viruses causing mild symptoms, *e.g.* TRV or TYMV, i) produce VSRs, 16K or P69, which partially inhibit the production of secondary siRNAs, and ii) induces the expression of *RTL1* to further reduce the production of secondary siRNAs, thus reinforcing the action of the VSRs.

Altogether, our study reinforces the idea that inhibiting PTGS amplification but not PTGS execution is a strategy commonly used by viruses to limit the plant PTGS defense and propagate without killing their host. Indeed, at least in the case of TRV or TYMV infection, the remaining PTGS activity due to primary siRNAs allows infected plants to survive, flower and produce seeds and thus transmit the virus. In the case of TYMV, a fraction of the seeds of an Arabidopsis infected plant transmit the virus [34], which thus spread all around the original site of infection as the newly infected plants and become a new source for TYMV propagation to other host plants. Therefore, it is tempting to speculate that the dual action of P69 and RTL1 on PTGS results in a tight balance between virus propagation and plant development. In the light of these results, one could consider the Arabidopsis-TYMV interaction as an elegant model of plant-virus coevolution.

## Materials and methods

### Plant material

Wild-type plants, transgenic lines, and loss-of-function mutants used in this study are in Arabidopsis Columbia (Col-0) ecotype or result from at least six back-crosses to Col-0.

The *6b4* line carries a *p35S:GUS* transgene that is stably expressed [23]. The *6b4 ski3*, *6b4 xrn4* and *6b4 vcs* lines exhibit S-PTGS due to dysfunctional RQC [20–22]. The *6b4-306* line exhibits IR-PTGS induced by the *p35S:hpG* construct expressing an hairpin consisting of the first half of the *GUS* sequence [23].

The *ago1-27*, *dcl2-1* (SALK_064627), *dcl4-2* (GABI_160G05), *dcl2-1 dcl4-2*, *rdr1-1* (SAIL_672F1), *rdr6* (*sgs2-1*) and *rdr1-1 rdr6* single and double mutants have been described previously [14, 24, 40–44]. The *ago2-3* (RATM15_3703) from the RIKEN collection was obtained from ABRC. *ago2-3* originally in the Nossen (No-0) ecotype was back-crossed six times to Col-0 for this study. The *ago1-27 ago2-3* double mutant was generated by standard crosses using *ago2-3* back-crossed to Col-0.

The T-DNA insertion mutant *rtl1-1* (SAILseq_337_F04.1) from the Syngenta Arabidopsis Insertion Library collection was obtained from the NASC. The CRISPR-Cas9 technology was used to generate *rtl1-2* mutant. A guide RNA targeting the 5' end of the *RTL1* (At4g15417) coding sequence was obtained using the CRISPOR website (http://crispor. tefor.net), synthesized by IDT (https://eu.idtdna.com) and cloned into the *pDE-Cas9-GentR* vector (Gateway Technology–Invitrogen/Thermo Fisher Scientific). Guide RNA and *rtl1-2* mutant sequence are described in S2 Fig. The *p35S:RTL1* plants have been previously described [19].

The *pPAB2:PAB2-RFP* line has been previously described [32]. The transgenic lines expressing *p35S:P69* were obtained using the *p35S:P69* construct previously described [18]. The transgenic lines expressing *p35S:SGS3-mCherry*, *pUBQ10:P69*, *pUBQ10:P69-GFP*, *pUBQ10:GFP-P69* or *pUBQ10:16K* were obtained as described below.

## Cloning and transformation

To generate the *pUBQ10:P69*, *pUBQ10:P69-GFP*, *pUBQ10:GFP-P69* constructs, the TYMV *ORF2* (*P69*) sequence was PCR-amplified from *pTY* [45] plasmids, using attb-flanked primers. A C-to-T mutation was introduced by PCR in TYMV *ORF2* at the 9th nucleotide to disrupt TYMV *ORF1* start codon. Sequences were then cloned into the *pDONR207* vector (Gateway Technology–Invitrogen/Thermo Fisher Scientific) and a LR reaction was performed with the *pUB-Dest* vector [46] to generate *pUBQ10:P69*. For *P69* subcellular localization studies, a LR reaction was performed with *pUBC-GFP-Dest* and *pUBN-GFP-Dest* vectors [46]. Primers used for cloning are listed in S1 Table.

To generate the *p35S:SGS3-mCherry* construct, LR reaction was performed with the *pMDC140-mCherry* vector using a *SGS3* clone in *pDONR207* vector [28]. The *p35S: SGS3-mCherry* construct was introduced into the *sgs3-1* mutant and lines showing complementation of the *sgs3* mutation were retained for confocal analysis.

Expression vectors were transferred into *Agrobacterium tumefaciens* C58C1 (pMP90) from *Escherichia coli* DH10B or DH5α bacteria (Thermo Fisher Scientific), either by electroporation or triparental mating, and Arabidopsis plants were transformed by floral dipping using an infiltration solution (5% sucrose, 10mM MgCl$_2$, 0.015% SilwetL-77) with Agrobacterium carrying the construct of interest at a final OD$_{600}$ of 1 [47]. Stable transformants were selected on medium supplemented with the corresponding antibiotics.

For agroinfiltration experiments, *N. benthamiana* leaves were infiltrated as described in [48] using an agroinfiltration solution (pH 5.2, 10mM MgCl$_2$, 10mM MES, 150mM acetosyringone) with Agrobacterium carrying the construct(s) of interest at a final OD$_{600}$ of 1. The *p35S:hpG* and *p35S:GFP* constructs used for this assay have been previously described [19, 23]. Leaves were harvested 3 days post-agroinfiltration for analysis.

## Growth condition and virus inoculation

Surface-sterilized seeds were sown *in vitro* on a Bouturage media (pH 5.9, 1.3% S-Medium S0262.0010, 1% Phyto Agar P1003, Duchefa Biochimie), vernalized at 4˚C for 48h and transferred in standard long-day conditions (16 hours light, 8 hours dark at 22˚C and 65% relative humidity). For subcellular localization assays, seeds were sown on vertical plates and roots were analyzed 5 days post-germination. For GUS analysis, two-week-old plantlets were transferred to soil in greenhouse with standard long-day conditions (16 hours day, 8 hours dark at ~22˚C and 45–60% relative humidity).

For virus infection assays, plants were grown directly on soil in controlled growth chamber in standard short-day conditions (8h of light, 16 hours of dark at 21˚C and 65% relative humidity).

For infection with *Turnip yellow mosaic virus*, four-week-old plants were infected by mechanical rubbing with carborundum powder and an *inoculum* of previously TYMV-infected *A. thaliana* leaves grinded in a 5mM $Na_2HPO_4$ 5mM $NaH_2PO_4$ buffer. For infection with *Tobacco rattle virus*, four-week-old plants were agroinfiltrated with a mix of agrobacteria carrying the *pTRV1* (*YL192*) and *pTRV2-MCS* (*YL156*) vectors [49] at a final $OD_{600}$ of 1. Rosettes of 16 plants were harvested 2 to 4 weeks post-inoculation for analysis.

## Quantification of viral symptoms

The projected rosette area per plant was measured with ImageJ on 4 plants per condition per genotype (manual measurement). Rosettes were segmented using Image>Adjust>Color threshold (Color space: Lab, Dark background, adjustment of the a* value). A binary image of the rosettes was obtained with Process>Binary>Make binary. The rosette area was selected with the wand tracing tool and measured with Analyze>Measure. Mean pixel area and standard deviation (error bar) were calculated.

## GUS assay

GUS activity was measured as described before [48]. Briefly, leaves grinded in a phosphate buffer (pH 7.2, 50 mM Na2HPO4, 50 mM $NaH_2PO_4$, 10 mM EDTA) are centrifuged for 20min at 4˚C and 3000 rpm. A Bradford protein assay was performed (Protein Assay Dye Reagent 500–0006, BioRad) with a BSA range and protein concentration was quantified with a ELx808 microplate reader (Biotek). Enzymatic activity was measured via the derived products generated from a 4-MUG substrate (M1404, Duchefa) with a Fluoroskan Ascent II (Thermo Fisher Scientific). GUS activity is presented in an arbitrary unit as the ratio between fluorescence data per minute and protein concentration.

## RNA analysis

RNA extraction and hybridization were performed as previously described [50]. Briefly, frozen leaves were grinded in liquid nitrogen, added to a NaCl extraction buffer (0.1M NaCl, 2% SDS, 50 mM Tris/HCl pH 9, 10 mM EDTA pH 8, 20 mM β-mercaptoethanol) and total RNA was extracted using a standard Phenol-Chloroform procedure. RNA was recovered in a 3v of 100% EtOH and 1/10v 3M NaOAc (pH 5.2) buffer at -80˚C for 1 hour. After a series of centrifugation, RNA pellets were resuspended in sterile water and quantified with a NanoDrop 2000C (Ozyme). For Low Molecular Weight (LMW) northern blot, 5 to 30 ug of RNA were denatured at 85˚C for 5 min, separated on a 15% polyacrylamide, 7.5M urea and 1X TBE gel and transferred on a Hybond NX membrane (Amersham). For High Molecular Weight (HMW) northern blot, 5 ug of RNA were denatured at 85˚C for 5min, separated on a 0.8 to 1.5% agarose,

0.7% formaldehyde, 20 mM HEPES and 1 mM EDTA pH 7.8 gel and transferred on a Genescreen Plus membrane (NEF-976, NEN/DuPont). PCR-probes were produced using primers listed in S1 Table, purified with a NucleoSpin Gel & PCR Clean-up kit (Machery-Nagel) and radiolabeled dCTP-P$^{32}$ were incorporated with a Prime-a-gene Labeling System kit (U1100, Promega). Oligonucleotide probes were ordered from GenoScreen and radiolabeled dATP-P$^{32}$ were incorporated with a T4 Polynucleotide Kinase kit (T4 PNK EK0031, Thermo Fisher Scientific). After blot saturation with salmon sperm, hybridization was performed in a PerfectHyb buffer (H7033, Sigma-Aldrich) overnight at 37˚C (dCTP-P$^{32}$ labeled probes) or at 50˚C (dATP-P$^{32}$ labeled probes) for LMW northern blot and in a Church buffer (7% SDS, 250 mM Na$_2$HPO$_4$, 2mM EDTA 200 μg/mL Heparin) overnight at 65˚C for HMW northern blot. After exposition on a BAS-MP 2040P imaging plate (Fujifilm), hybridization signal was revealed with a Typhoon-FLA9500 phosphoimager (Ge-Healthcare). RNA band intensity was measured on unsaturated image with ImageJ. Data were normalized to the band intensity of the loading control.

For TYMV quantitative RT-PCR analysis, the quantification method described before [51] was followed to estimate virus accumulation in TYMV-infected tissue. A pTY plasmid [45] was linearized at a unique HindIII restriction site and plasmid copy number was estimated using the Avogadro's constant. A qRT-PCR was performed on a tenfold nine-points dilution series with 3 technical replicates to calculate the standard curve formula y = -2.073x + 7.4178. A qRT-PCR was performed on TYMV-infected leaves as described above except for random primers used instead of oligo dT for cDNA synthesis and data are analyzed with the standard curve to estimate TYMV quantity. Primers are listed in S1 Table and qRT-PCR cycle conditions are listed in S2 Table.

## Subcellular localization experiments

GFP, mCherry and RFP signal on epidermis of *N. benthamiana* agroinfiltrated leaves or roots of five-day-old *A. thaliana* seedlings were analyzed with a Leica TSC SP5 confocal and water-immersion objectives before and/or after a heat stress at 37˚C for one hour. Images were analyzed with ImageJ.

## Supporting information

**S1 Fig. The plant PTGS defense limits TYMV infection through the action of AGO1, AGO2 and DCL4.** Viral gRNA accumulation in *Arabidopsis thaliana* wild type and PTGS mutant plants four weeks after infection with TYMV. Total RNA was extracted from a pool of 16 infected plants per genotype. The amount of TYMV gRNA was quantified by qRT-PCR using primers located on the TYMV ORF2 P69. *GAPDH* was used as intern control. Data are normalized to Col.
(TIFF)

**S2 Fig. Description of the *rtl1-2* mutant.** A guide RNA targeting *RTL1* (AT4G15417) sequence (indicated in yellow) was synthesized as described in the Materials and Methods. Sanger sequencing was performed to identified mutations and a BLAST analysis was conducted with SerialCloner. A T is introduced at the 27$^{th}$ position (arrow in red) 3-nt before the PAM site (indicated in blue). A premature stop-codon appeared at the 8$^{th}$ amino acid and a second TasI restriction site at the insertion, allowing genotyping with primers indicated in green. Segregation of the *Cas9* cassette was followed by PCR and loss of antibiotic resistance. *rtl1-2* was back-crossed six times to Col before analysis.
(TIFF)

**S1 Table. Primers used for genotyping of newly described mutant, probes synthesis and hybridization.**
(XLSX)

**S2 Table. qRT-PCR programs.**
(XLSX)

## Acknowledgments

We thank Cécile Antonelli for the *pPAB2:PAB2-RFP* line, Shou-Wei Ding for the *p35S:P69* plasmid, and Isabelle Jupin for the *TYMV* plasmid. This work has benefited from the support of IJPB's Plant Observatory technological platforms. The IJPB benefits from the support of Saclay Plant Sciences-SPS (ANR-17- EUR-0007). This work has benefited from a French State grant (Saclay Plant Sciences, reference ANR-17-EUR-0007, EUR SPS-GSR) managed by the French National Research Agency under an Investments for the Future program (reference ANR-11- IDEX-0003-02).

## Author Contributions

**Conceptualization:** Hayat Sehki, Taline Elmayan, Hervé Vaucheret.

**Formal analysis:** Hayat Sehki, Agnès Yu, Taline Elmayan, Hervé Vaucheret.

**Funding acquisition:** Hervé Vaucheret.

**Investigation:** Hayat Sehki, Taline Elmayan, Hervé Vaucheret.

**Methodology:** Hayat Sehki, Agnès Yu, Taline Elmayan, Hervé Vaucheret.

**Project administration:** Hervé Vaucheret.

**Resources:** Hervé Vaucheret.

**Supervision:** Hervé Vaucheret.

**Writing – original draft:** Hayat Sehki.

**Writing – review & editing:** Hayat Sehki, Taline Elmayan, Hervé Vaucheret.

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
