## [Decision Letter · Decision Letter 0]

13 May 2022

Dear Prof. Hervé Vaucheret,

Thank you very much for submitting your manuscript "TYMV has dual action on the plant RNA silencing defense through its VSR P69 and the host RNASE THREE LIKE1" (PPATHOGENS-D-22-00563) for review by PLoS Pathogens. Your manuscript was fully evaluated at the editorial level and by independent peer reviewers. Given the reviewers’ comments, we would like to invite you to prepare and submit a revised manuscript.

As you will see, the reviewers overall appreciate that this manuscript provides some interesting findings to the antiviral silencing field. However, they do raise some points that should be addressed in the revised version. In particular, the inoculated plants in Figure 1 should be in good shape and need mock-treated controls. The qPCR results of Fig 1B and 6D need enough independent repeats to analyze error bars and significance test. In addition, TYMV infection symptom images of rtl mutants and viral RNA accumulation data should be provided. These issues must be addressed before we would be willing to consider a revised version of your study. We therefore ask you to modify the manuscript according to the review recommendations before we can consider your manuscript for acceptance.

(1) A letter containing a detailed list of your responses to the review comments and a description of the changes you have made in the manuscript.

(2) Two versions of the manuscript: one with either highlights or tracked changes denoting where the text has been changed; the other a clean version (uploaded as the manuscript file).

We hope to receive your revised manuscript within 90 days. If you anticipate any delay in its return, we ask that you let us know the expected resubmission date by replying to this email. Revised manuscripts received beyond 90 days may require evaluation and peer review similar to that applied to newly submitted manuscripts.

Sincerely,

Xian-Bing Wang

Guest Editor

PLOS Pathogens

Shou-Wei Ding

Section Editor

PLOS Pathogens

Kasturi Haldar

Editor-in-Chief

PLOS Pathogens

orcid.org/0000-0001-5065-158X

Grant McFadden

Editor-in-Chief

PLOS Pathogens

orcid.org/0000-0002-2556-3526

Reviewer's Responses to Questions

**Part I - Summary**

Reviewer #1: This manuscript presented interesting finding regarding the mechanism of silencing suppression by TYMV p69 and the Arabidopsis endogenous silencing suppressor RTL1. The authors first showed that Arabidopsis dcl4, ago1 and ago2 mutants but not dcl2, rdr1 and rdr6 mutants showed enhanced susceptibility to TYMV infection. Next the authors demonstrated p69 was capable of inhibit S-PTGS but not IR-induced PTGS which was consistent with previous report from another lab. It was further showed that p69 was colocalized with SGS3 in the siRNA body. Based these evidences the author proposed that p69 suppresses antiviral defense by inhibiting the RDR mediated amplification step. The manuscript also showed that mutation of rtl1 enhanced accumulation of TYMV siRNA and reduced level of its genomic RNAs which was complement to the previous report about the RTL1 overexpression phenotype. Though this manuscript provides some interesting findings to the antiviral silencing field. There are some major issues to be addressed to make a more solid ground for the conclusion to stand.

Reviewer #2: Manuscript PPATHOGENS-D-22-00563 reports on the mechanisms of silencing suppression by Turnip yellow mosaic virus (TYMV) in Arabidopsis thaliana. TYMV accumulation was enhanced in ago1, ago2 and dcl4 mutants, which are impaired in the execution of post-transcriptional silencing, but not in dcl2, rdr1 and rdr6 mutants, which are impaired in the amplification of post-transcriptional silencing. A genetic analysis showed that TYMV suppresses RNA silencing amplification by two mechanisms: P69-mediated suppression and by inducing the expression of the host enzyme RNASE THREE-LIKE 1 (RTL1)-mediated. Results contribute novel understanding of plant-virus interactions, and silencing suppression mechanisms. Minor adjustments are required for scientific accuracy.

MAIN CONTRIBUTIONS

- Compared to wt Arabidopsis, TYMV accumulation is higher in ago1, ago2 and dcl4 mutants, which are impaired in the execution of PTGS.

- Compared to wt Arabidopsis, TYMV accumulation is not enhanced in dcl2, rdr1 and rdr6 mutants, which are impaired in the amplification of PTGS.

- TYMV suppresses RNA silencing amplification by two mechanisms: P69-mediated suppression and by inducing the expression of the host enzyme RNASE THREE-LIKE 1 (RTL1)-mediated.

- TYMV P69 is a weak silencing suppressor that partially inhibits the amplification but not the execution of RNA silencing. P69 localizes in siRNA-bodies where it partially inhibits the production of secondary siRNAs.

- TYMV P69 does not inhibit RTL1 activity.

- TYMV-derived siRNAs are limited to the primary siRNAs, which limits the magnitude of the antiviral response.

- Symptoms are mild due to reduced amounts of siRNA/virus accumulation.

Reviewer #3: Small RNA-dependent RNA silencing pathway is a natural antiviral mechanism in plants, and viruses encodes suppressor proteins to inhibit host RNA silencing. However, a strong suppression of host RNA silencing will lead to an excessive accumulation of virus, which is bad for the long-terminal survival of the virus in host. How the viruses maintain the balance between successful infection and long-term survival is unclear. In this manuscript, Sehki et al demonstrated that TYMV impairs the amplification but not the execution of PTGS via VSR P69.They showed that P69 colocalizes in the siRNA-bodies with secondary siRNAs to inhibit PTGS amplification. In addition, they revealed that TYMV can induce the expression of plant RNASE THREE-LIE 1 (RTL1), thereby antagonizing siRNA accumulation. They concluded that the dual function of TYMV through VSR P69 and RTL1 is an elegant strategy for viruses to balance plant defense and viral propagation in host. I have to say, this finding is very interesting and worth study deeply. However, I think these hypotheses raised by the authors are not well supported by the data presented in the current manuscript. The overall content of the manuscript lacks a clear logical relationship, especially the two parts of P69 and RTL1 that seem completely separate. More evidence should be provided to support their hypothesis.

**Part II – Major Issues: Key Experiments Required for Acceptance**

Reviewer #1: Major issues:

1. The quality of viral infection data in Figure 1A should be improved. It looked like the plants were watered too much and not in good shape. And mock inoculated negative control should be shown. Biological repeats and statistics for analysis of viral RNA accumulation in different genotypes are lacking (Figure 1B).

2. One evidence indicating that p69 inhibit RDR6 amplification is that p69 expression in Arabidopsis inhibit TAS3 siRNAs (Figure 4 B). As TAS3 siRNAs are dependent on miR390. It is necessary to compare the level of miR390 in absence and presence of p69. If p69 indeed inhibit RDR6 amplification step, miR390 mediated cleavage product of TAS3 should be still detected in both 6b4-306 and 6b4-306/p69 plants. Similarly, cleavage products containing GUS central sequences should also be detected in both genotypes.

3. Based on the subnuclear localization experiments in Arabidopsis roots and Nicotiana benthemiana leaves, p69 seems localized in differently in different organ, which may indicate it may function differently in these tissues. Analysis of GUS activity and small RNA accumulation in Arabidopsis roots or analyze p69 subcellular localization in Arabidopsis leaves is suggested to make a better ground for a conclusion about p69 function mechanism.

4. It is clear that RTL1 negatively regulates antiviral silencing against TYMV but not very convincing to conclude that RTL1 function in corporation with p69 to degrade amplified viral siRNAs based on just single rtl1 mutant phenotype. Comparison of viral siRNAs between rdr1/6 and rdr1/6/rtl1 may help test if RTL1 function through RDR mediated amplification step. An Y2H is also suggested to test whether RTL1 and p69 interact with each other.

Reviewer #2: None.

Reviewer #3: 1. The writing of the manuscript is also very unsmooth. There are too many technical terms, which make me very difficult to read smoothly.

2. Fig 1A.The plants appear to have experienced severe environmental stress which may affect the disease symptom. In addition, an uninfected mock control is missing in all infection experiments.

3. To strength the authors’ conclusion that the inhibition of PTGS amplification depends on P69, I would suggest that authors to use a P69-deleted TYMV to perform virus infection assay in different RNA silencing mutants.

4. Fig 1B and Fig 6D. The qPCR results lack error analysis and significance test of the difference. Moreover, to more accurately evaluate the accumulation of viral RNAs in plants, more experiments like RNA blotting assay should be performed.

5. Fig 5B. To better support the colocalization of P69 with secondary siRNAs in siRNA bodies, a colocalization of P69 with RDR6 should be provided.

6. Fig 6. TYMV infection symptom images of rtl mutants and viral RNA accumulation data should be provided.

**Part III – Minor Issues: Editorial and Data Presentation Modifications**

Reviewer #1: Minor problem:

1. Figure 4 B “GUS centrql” should be “GUS central”.

2. Involvement of RTL1 in antiviral silencing is interesting, a picture of infected plants in comparison with wild type plants is suggested to show in the figure.

Reviewer #2: - RQC needs to be explained.

- Error bars are needed in Fig. 1B. This is critical, because the paragraph on lines 138 to 150 is based on comparisons between single and double mutants.

- Line 144. AGO proteins do not produce primary siRNAs.

- Virus family and species need to follow standard nomenclature. Example, Potyvirus family is inaccurate.

- Both P19 and P38 are assigned to the Tombusviridae. Clarification is needed.

Reviewer #3: Line 180, What is QRC, Please explain.

Fig 4B, The word “Centrql” should be “Central”.

PLOS authors have the option to publish the peer review history of their article (what does this mean?). If published, this will include your full peer review and any attached files.

Reviewer #1: No

Reviewer #2: **Yes: **Hernan Garcia-Ruiz

Reviewer #3: No
---

## [Decision Letter · Decision Letter 1]

10 Jan 2023

Dear Prof. Vaucheret

We are pleased to inform you that your manuscript 'TYMV and TRV infect Arabidopsis thaliana by expressing weak suppressors of RNA silencing and inducing host RNASE THREE LIKE1' has been provisionally accepted for publication in PLOS Pathogens.

Best regards,

XianBing Wang

Guest Editor

PLOS Pathogens

Shou-Wei Ding

Section Editor

PLOS Pathogens

Kasturi Haldar

Editor-in-Chief

PLOS Pathogens

orcid.org/0000-0001-5065-158X

Michael Malim

Editor-in-Chief

PLOS Pathogens

orcid.org/0000-0002-7699-2064

Reviewer Comments (if any, and for reference):

Reviewer's Responses to Questions

**Part I - Summary**

Reviewer #1: This manuscript presented interesting finding regarding the mechanism of silencing suppression by TYMV and TRV, which encode mild viral suppressor of RNA silencing by targeting amplification of silencing and thus require induction of the Arabidopsis endogenous silencing suppressor RTL1 to suppress antiviral defense coopperatively. Data quality in this revision has substaintially improved and my concerns are very well addressed.

Reviewer #3: In this revision, most of my major concerns have been solved. However, I still cannot understand the authors' explanation that they cannot obtain a full set of RNAs of sufficiently good quality for either qRT-PCR or northern blot analysis. This explanation is unreasonable because these mentioned experiments are very general assays in plant-virus interaction field. In addition, the resolution of Figure 1B and Figure 6B are quite low, which make me very hard to make any evalution about the data.

**Part II – Major Issues: Key Experiments Required for Acceptance**

Reviewer #1: (No Response)

Reviewer #3: (No Response)

**Part III – Minor Issues: Editorial and Data Presentation Modifications**

Reviewer #1: The characters in labeling growth curves in Figure 1 and Figure 6 B are too small and needs to be adjusted.

Reviewer #3: (No Response)

PLOS authors have the option to publish the peer review history of their article (what does this mean?). If published, this will include your full peer review and any attached files.

Reviewer #1: No

Reviewer #3: No

---

## [Editor Report · Acceptance letter]

20 Jan 2023

Dear Dr. Vaucheret,

We are delighted to inform you that your manuscript, "TYMV and TRV infect Arabidopsis thaliana by expressing weak suppressors of RNA silencing and inducing host RNASE THREE LIKE1," has been formally accepted for publication in PLOS Pathogens.

Best regards,

Kasturi Haldar

Editor-in-Chief

PLOS Pathogens

orcid.org/0000-0001-5065-158X

Michael Malim

Editor-in-Chief

PLOS Pathogens

orcid.org/0000-0002-7699-2064